# The Role of PIVKA-II as a Predictor of Early Hepatocellular Carcinoma Recurrence-Free Survival after Liver Transplantation in a Low Alpha-Fetoprotein Population

**DOI:** 10.3390/cancers16010004

**Published:** 2023-12-19

**Authors:** Monique J. C. Devillers, Johanna K. F. Pluimers, Maria C. van Hooff, Michail Doukas, Wojciech G. Polak, Robert A. de Man, Milan J. Sonneveld, Andre Boonstra, Caroline M. den Hoed

**Affiliations:** 1Department of Gastroenterology and Hepatology, Erasmus MC Transplant Institute, Erasmus MC University Medical Center, P.O. Box 2040, 3000 CA Rotterdam, The Netherlands; m.devillers@erasmusmc.nl (M.J.C.D.); j.pluimers@erasmusmc.nl (J.K.F.P.); m.vanhooff@erasmusmc.nl (M.C.v.H.); r.deman@erasmusmc.nl (R.A.d.M.); m.j.sonneveld@erasmusmc.nl (M.J.S.); p.a.boonstra@erasmusmc.nl (A.B.); 2Department of Pathology, Erasmus MC University Medical Center, 3000 CA Rotterdam, The Netherlands; m.doukas@erasmusmc.nl; 3Department of Surgery, Division of Hepatopancreatobiliary and Transplant Surgery, Erasmus MC Transplant Institute, Erasmus MC University Medical Center, 3000 CA Rotterdam, The Netherlands; w.polak@erasmusmc.nl

**Keywords:** hepatocellular carcinoma, liver transplantation, recurrence, DCP, PIVKA-II, risk score

## Abstract

**Simple Summary:**

Although the RETREAT score improves hepatocellular carcinoma (HCC) recurrence prediction post liver transplantation (LT), there is a necessity for more discriminating models in low alpha-fetoprotein (AFP) populations. Investigating the use of a protein induced by vitamin K deficiency or antagonist-II (PIVKA-II) as a predictor after liver transplantation, this study shows that patients with low PIVKA-II in combination with low AFP or low PIVKA-II within the Milan criteria have 100% recurrence-free survival. This suggests that these groups might not need HCC surveillance after LT. PIVKA-II might outperform AFP in predicting microvascular invasion on explant and could assist in identifying LT candidates with the highest risk of HCC recurrence in future models.

**Abstract:**

Introduction: AFP and the RETREAT score are currently used to predict HCC recurrence after LT. However, superior discriminating models are needed for low AFP populations. The aim of this study is to investigate the predictive value of PIVKA-II on recurrence-free survival after LT in a low AFP population and microvascular invasion on explant. Methods: A retrospective cohort study including all consecutive patients transplanted for HCC between 1989 and 2019 in the Erasmus MC University Medical Center in Rotterdam, the Netherlands, was used. AFP and PIVKA-II levels were determined in serum samples collected at the time of transplantation. Data on tumor load and microvascular invasion were retrieved from patients’ records. Results: The study cohort consisted of 121 patients, with HCC recurrence in 15 patients (12.4%). The median AFP was 7.7 ng/mL (4.4–20.2), and the median PIVKA-II was 72.0 mAU/mL (41.0–213.5). Patients with low AFP (≤8 ng/mL) and PIVKA-II (≤90 mAU/mL) had a 5-year recurrence-free survival of 100% compared to 85.7% in patients with low AFP and high PIVKA-II (*p* = 0.026). Regardless of the AFP level, patients within the Milan criteria (based on explant pathology) with a low PIVKA-II level had a 5-year recurrence-free survival of 100% compared to patients with a high PIVKA-II level of 81.1% (*p* = 0.002). In patients with microvascular invasion, the AUC for PIVKA-II was slightly better than the AUC for AFP (0.775 vs. 0.687). Conclusions: The dual model of PIVKA-II ≤ 90 mAU/mL with either AFP ≤ 8 ng/mL or with patients within the Milan criteria identifies patient groups which can be exempted from HCC surveillance after LT in a low AFP population. PIVKA-II may be a better predictor for explant microvascular invasion than AFP and could play a role in future models identifying LT candidates with the highest risk for HCC recurrence.

## 1. Introduction

The best curative treatment for patients with early stage hepatocellular carcinoma (HCC) is surgical resection. However, most patients develop HCC in the context of an underlying liver disease for which liver transplantation (LT) can be a superior treatment. Due to a scarcity of organs, there is a need for a strict selection of patients that will benefit from LT. Despite stringent selection criteria, the recurrence of HCC occurs in 8–20% of the patients after LT [1].

The majority of recurrences occur during the first two years after LT (75%) with a median survival after recurrence of 7 to 16 months [2,3]. Currently, there is no consensus on monitoring for tumor recurrence after LT, and there is a wide variation in follow-up regimens internationally [4]. The determination of the risk of recurrence could contribute to a more fitting surveillance strategy for patients at risk of recurrence. Some patients could benefit from early detection and local treatment, possibly prolonging their survival. [5]. To predict HCC recurrence after LT, Mehta et al. (2017) developed the RETREAT score. The score is assessed at time of LT and derives from the sum of the largest viable tumor diameter and the number of viable tumors on explant, microvascular invasion on explant and alpha-fetoprotein levels (AFP) [6].

Recently, the RETREAT score was validated in our cohort of patients who underwent LT for HCC at the Erasmus University Medical Center, Rotterdam, the Netherlands [7]. Since the average AFP in our population is low, the addition of other biomarkers or characteristics might strengthen the predictive value of the RETREAT score in our population [3,8]. Furthermore, a surrogate marker that can predict microvascular invasion prior to LT could identify LT candidates at high risk of HCC recurrence.

One of the biomarkers of interest is protein induced by vitamin K deficiency or antagonist-II (PIVKA-II), also known as des-gamma carboxyprothrombin (DCP). PIVKA-II derives from the prothrombin precursor in HCC cells, in which glutamate residues are post-translationally carboxylated to form γ-glutamic acid. [9]. Therefore, PIVKA-II is an abnormal type of prothrombin secreted by HCC cells and has no coagulation function [10]. Several studies have demonstrated that PIVKA-II is a superior biomarker for the diagnosis of HCC compared to AFP [11,12] and that PIVKA-II is a predictor of HCC recurrence after curative resection [13]. Limited but promising research on the predictive value of PIVKA-II for HCC recurrence after LT has been conducted [8,14,15,16].

However, these studies have been performed in Eastern populations and have not been confirmed in a Western population with predominantly low AFP levels. The aim of this study is to compare HCC recurrence-free survival probabilities in a low AFP population with low or high PIVKA-II levels stratified according to low or high RETREAT scores and whether PIVKA-II can be used as a predictor for explant microvascular invasion.

## 2. Methods

### 2.1. Patients

In this study, all consecutive HCC patients receiving LT between October 1989 and October 2019 in our center were included, and the data were retrospectively evaluated. For a more detailed description of this cohort, see Van Hooff et al. [7]. Before LT, all patients provided written informed consent, approving the use of their data and biomaterial for research purposes. The current study was approved by the medical ethical committee of the Erasmus MC (MEC-2019-0775) [7]. Follow-up data were collected until November 2023.

### 2.2. Pre-Transplant Data

Pre-transplant data were collected from patient records and included age, sex, underlying liver disease, time on the waiting list, locoregional therapy and therapy with curative intent before transplantation. Treatment with curative intent before transplantation consisted of liver resection or ablation (either radio frequency ablation (RFA) or microwave ablation (MWA)). Locoregional therapy, also known as downstaging or bridging therapy, comprised transarterial chemoembolization (TACE), transarterial radioembolization (TARE), percutaneous ethanol injection (PEI) or stereotactic radiotherapy (SRT). Bilirubin, creatinine, sodium and INR, on the day of transplantation, were retrieved and used to calculate the laboratory MELD score.

### 2.3. Post-Transplant Follow-up and Diagnosis of HCC Recurrence

As described previously, a follow-up of LT patients consisted of regular clinical visits, monitoring AFP, biannual ultrasounds, a yearly X-ray of the chest and an electrocardiogram. Further imaging was only performed whenever clinical symptoms or laboratory abnormalities were detected or according to the physicians’ judgement. In case recurrence was suspected clinically, additional comprehensive CT imaging was performed. The diagnosis of HCC recurrence was based on histopathology reports or clinical consensus [7]. The last follow-up date was defined as the last clinical visit with no evidence of HCC recurrence on imaging or in laboratory tests including AFP levels.

After the RETREAT score was validated in our cohort in 2020 [7], the surveillance strategy was updated accordingly. In patients with a RETREAT score of 3 or higher, a thoracoabdominal CT scan was performed, and AFP-levels were determined every 6 months. Patients with a RETREAT score under 3 received a yearly follow-up with a thoracoabdominal CT and AFP levels. If AFP levels were elevated by more than 1× above the normal range, or if symptoms were present, a bone scan was conducted. This prospective follow-up ended 5 years after LT.

### 2.4. Biomarker Determination

Pre-transplantation-collected serum samples were gathered from the biobank of the Erasmus Medical Centre. All of the available serum samples were collected on the day before or on the day of transplantation. The serum samples had been stored after collection in a −20 degrees Celsius freezer. Patients were excluded whenever there was not enough archived serum.

After the collection of the serum samples, the samples were thawed, and an amount of 380 µL was collected for analysis. AFP and PIVKA-II were determined using the automated immunoassay instrument LUMIPULSE G1200 (Fujirebio, Tokyo, Japan). The instructions of the manufacturer were followed to perform the assays. Serum AFP levels were assessed in ng/mL with the LUMIPULSE G AFP-N kit, with low and high limits of detection of 0.5 and 2000 ng-mL, respectively. Serum PIVKA-II levels were assessed in mAU/mL with the LUMIPULSE G PIVKA-II kit (Fujirebio Inc., Gent, Belgium) with low and high levels of 5 and 75.000 mAU/mL, respectively. A grid search was performed to determine the cutoff values of the biomarkers for the purpose of identifying the patients with the lowest recurrence risk. Patients using a vitamin K antagonist at the day of transplantation were excluded, because the serum levels of PIVKA-II are found to be higher in patients using a vitamin K antagonist [15,16].

### 2.5. Histopathology

All liver explants were assessed by an expert pathologist. The histopathology reports of the liver explants were reviewed to determine the following variables: the presence of microvascular invasion, a tumor differentiation grade, the number of viable tumor(s) and the diameters of viable tumor(s). The differentiation grade of the tumor was categorized as well, moderately or poorly differentiated based on the criteria of the World Health organization (WHO) [17]. Tumors were categorized as no viable tumor if there was no HCC on the explant. In case of multiple tumors showing different differentiation grades, the worst differentiation grade was noted.

### 2.6. Milan Criteria on Explant and RETREAT Score

Based on histopathology, all patients were categorized as within or exceeding the Milan criteria. The RETREAT score was calculated utilizing pre-transplant AFP levels and histopathological features, as previously described [6]. The AFP value used was determined in the serum samples collected at the day of transplantation. A patient’s RETREAT score could range from 0 to a maximum of 8 points.

### 2.7. Statistical Analysis

Statistical analyses were performed using IBM SPSS Statistics version 25 and GraphPad Prism version 9.2.0. Categorical variables were represented as a proportion and compared using a chi-squared test or Fisher’s exact tested where appropriate. Continuous variables were represented as a mean (standard deviation) and tested using the independent samples *t*-test or a median (interquartile range) and tested using the Mann–Whitney U nonparametric test.

The primary outcome was the ability of PIVKA-II to predict time to recurrence (TTR) of HCC in patients with low AFP levels and in patients within the Milan criteria on explant. A TTR is a Kaplan–Meijer curve in which recurrence is the single event and death is used as a censoring point but is not regarded as an event [18]. The secondary outcome was the comparison of TTR probabilities in patients with low or high PIVKA-II levels stratified according to low or high RETREAT scores. Throughout this study, TTR probabilities will be referred to as recurrence-free survival. Lastly, the ability of PIVKA-II to predict explant tumor microvascular invasion pre-transplantation was assessed.

Recurrence-free survival curves were computed using the Kaplan–Meier method and compared using the log-rank test. The level of significance was set at *p* < 0.05, and all hypotheses were two-tailed tested. A RETREAT score of 0–2 was categorized as low, and a score of 3 or more was categorized as high, as previously described (8). The ability of PIVKA-II to predict tumor microvascular invasion was quantified with the area under the receiver-operating characteristic curves.

## 3. Results

Out of 216 patients that underwent LT because of HCC between 1989 and 2019, 121 patients were included in this study (Figure 1). The remaining 95 were excluded because of missing data (80), the use of vitamin K antagonists (7), the presence of needle tract seeding metastasis (4), or they were lost to follow-up (4).

### 3.1. Patient Characteristics

Patient and tumor characteristics are summarized in Table 1. The cohort was predominantly male with a median age of 60.7 years. Viral hepatitis was the most common underlying liver disease, with hepatitis C virus in 23 patients (18.9%) and hepatitis B virus in 18 patients (14.8%). The remainder (*n* = 12) had a combined viral etiology. Almost half of the cohort (49.2%) received treatment with curative intent (RFA, MWA or liver resection) before transplantation. Over a quarter (26.4%) received locoregional therapy (TACE, TARE, ILC, PEI or stereotactic radiotherapy) as downstaging before transplantation. Despite the selection criteria for LT, 20.7% patients exceeded the explant histopathology of the Milan criteria. Microvascular invasion was present in 31 cases (25.6%). The median follow-up time of the total cohort was 65.9 months (IQR 37.6–105.0).

### 3.2. HCC Recurrence

HCC recurrence occurred in 15 patients (12.4%), with a median time between LT and diagnosis of HCC recurrence of 18.2 months (8.0–30.4). A significantly larger proportion of patients with HCC recurrence was outside of the Milan criteria based on explant histopathology compared to the patients without HCC recurrence, 53.3% vs. 16.0% (*p* = 0.003), respectively. Patients with HCC recurrence showed a significantly larger tumor bulk, a higher incidence of microvascular invasion and, more often, a more poorly differentiated tumor (*p* < 0.001, *p* < 0.001 and *p* = 0.021, respectively).

### 3.3. PIVKA-II in Addition to AFP for HCC Recurrence and Microvascular Invasion

The median values of AFP and PIVKA-II were 7.7 ng/mL (IQR 4.4–20.2) and 72.0 mAU/mL (IQR 41.0–213.5), respectively. AFP and PIVKA-II were significantly lower in patients without HCC recurrence compared to patients with HCC recurrence, 6.4 ng/mL (IQR 4.3–14.7) vs. 25.2 ng/mL (IQR 8.5–1284.4, *p* < 0.001), and 68.0 mAU/mL (IQR 37.8–163.5) vs. 217.0 mAU/mL (IQR 79.0–836.0, *p* = 0.005), respectively. The distribution of the AFP and PIVKA-II levels for the patients stratified by HCC recurrence, the Milan criteria and microvascular invasion are displayed in Figure 2.

When stratifying the patients by the Milan criteria, patients within the Milan criteria who did not develop HCC recurrence had significantly lower PIVKA-II levels than patients with HCC recurrence, 65.0 mAU/mL (IQR 37.0–154.0) vs. 217.0 mAU/mL (IQR 184.4–879.0, *p* = 0.003) (Table 2). Similarly, patients without microvascular invasion at explant had significantly lower PIVKA-II levels.

### 3.4. Recurrence-Free Survival Curves in Low AFP Level Patients and in Patients within the Milan Criteria

Patients were categorized as high or low AFP and PIVKA-II based on a cutoff level of 8.0 ng/mL and 90.0 mAU/mL, respectively, as determined by the grid search. The 5-year cumulative recurrence-free survival in the low AFP group (≤8.0 ng/mL) was 95.2% compared to 79.3% in patients with high AFP (>8.0 ng/mL) (*p* = 0.017). Patients with both low AFP and low PIVKA-II had a 5-year cumulative recurrence-free survival of 100% compared to 85.7% in patients with low AFP and high PIVKA-II (*p* = 0.026; Figure 3A). Regardless of the AFP level, patients within the Milan criteria with a low PIVKA-II level had a 5-year cumulative recurrence-free survival of 100% compared to patients with a high PIVKA-II level of 81.1% (*p* = 0.002; Figure 3B).

### 3.5. Modifying the RETREAT Score with PIVKA-II

In the total study cohort, we identified 81 (66.9%) patients with a low RETREAT score (0–2) with 1 recurrence and 40 (33.1%) patients with a high RETREAT score (≥3) and 14 recurrences. The 1-, 3- and 5-year cumulative recurrence-free survival in the high RETREAT score group was 83.8%, 70.1% and 60.8%, respectively. We identified 40 (33.0%) patients with a high RETREAT score (≥3), who were stratified in a low (*n* = 10) and high PIVKA-II group (*n* = 30). Regarding patients with a high RETREAT score, the 1-, 3- and 5-year cumulative recurrence-free survival in patients with a low PIVKA-II level were 90.0%, 75.0% and 75.0%, compared to 81.5%, 63.0% and 55.3% in the high PIVKA-II group (*p* = 0.248), respectively. In the patient group with a low RETREAT score (0–2), 56 patients were identified with a low PIVKA-II level and 25 patients with a high PIVKA-II level. There was one HCC recurrence in this group.

### 3.6. PIVKA-II and Microvascular Invasion

Patients were divided into four groups according to high or low AFP and PIVKA-II (Table 3). In patients with microvascular invasion, 48.4% had either a high AFP or high PIVKA-II, and 51.6% had a high AFP and PIVKA-II (*p* < 0.001). No microvascular invasion was seen in the group of patients with low AFP and low PIVKA-II. Importantly, the AUC for PIVKA-II is higher than the AUC for AFP (0.775 vs. 0.687, respectively) (Figure 4).

## 4. Discussion

Despite the use of allocation models such as the Milan criteria, HCC recurrence after transplantation occurs in a significant proportion of patients. Currently, there is no prognostic scoring system for HCC recurrence after LT that is reliable enough to be the foundation of a thorough surveillance strategy. In 2017, Mehta et al. developed the RETREAT score which is based on tumor number and diameters, microvascular invasion, and the biologic tumor marker AFP [6]. The RETREAT score has been externally validated [7,19,20,21] even in a population with low average AFP levels [7]. In addition to AFP, other discriminating biomarkers used to predict HCC recurrence have been identified. The use of the biomarker PIVKA-II been shown to be useful in HCC diagnosis, as well as predicting HCC recurrence and predicting high-risk explant pathology in a dual biomarker model [22]. In this single-center retrospective cohort study, we focused on the ability of PIVKA-II as a biomarker to predict recurrence-free survival in a population with low AFP levels. Our secondary objective was to evaluate PIVKA-II’s ability to predict explant microvascular invasion.

Previous studies predominantly focused on PIVKA-II’s ability to predict the occurrence of HCC recurrence post-transplant [13,15,22,23,24,25,26,27]. Our study is the first to identify patients with a 100% 5-year cumulative recurrence-free survival after transplantation in patients. This was observed in patients with a low PIVKA-II level in combination with either a low AFP (≤8.0 ng/mL) or within the Milan criteria. Two other comparable Korean studies found low recurrence rates in patients with low AFP and PIVKA-II levels. Kim et al. observed a 5-year recurrence-free survival of 91.9% in patients with an AFP cutoff level < 150 ng/mL and PIVKA-II level < 100 mAU/mL [14]. Park et al. demonstrated that patients with an AFP cutoff value < 20 ng/mL and PIVKA-II < 40 mAU/mL had a crude recurrence rate of 6.8% [15]. The only comparable study conducted in a Western population combined PIVKA-II with *Lens culinaris* aglutinen, a fycosylated form of AFP of the total AFP level, termed AFP-L3%. In the dual model, the 3-year recurrence-free survival was 97.0% with cutoff levels of AFP-L3 < 15% and PIVKA-II < 7.5 ng/mL [22]. Therefore, it seems that a low PIVKA-II level is able to further stratify patients with a low recurrence risk, identifying a group of patients that can be exempt from strict screening for HCC recurrence after liver transplantation.

In order to establish accurate AFP and PIVKA-II cutoff levels to predict high cumulative recurrence-free survival probabilities, future research should integrate different biomarker combinations at diverse cutoff levels across patients with different LT indications stratified by initial the Milan Criteria. Similarly, in future research, PIVKA-II with multiple cutoffs should be added to the RETREAT score into a scoring model and validated in a prospective study.

As mentioned previously, one of the most considerable limitations of the RETREAT score is that it can only be calculated after LT [7]. Given the scarcity of donor organs, the prediction of HCC recurrence prior to LT is of the utmost importance to select high-risk patients for LT exclusion. Our results demonstrated that PIVKA-II is a slightly better predictor of explant microvascular invasion and is useful in combination with AFP to identify the patients with microvascular invasion. This is in line with previous studies showing that singular PIVKA-II or PIVKA-II in combination with AFP-L3 is a stronger predictor of microvascular invasion than AFP in patients undergoing liver resection [22,28,29]. To identify high-risk candidates pre-emptive of LT, future prospective studies are needed that explore the predictive value of PIVKA-II at multiple cutoff levels, potentially in combination with other biomarkers such as AFP-L3, to accurately predict the presence of explant microvascular invasion.

A limitation of this study is the single center and retrospective design with a relative small sample size with a limited number of recurrences, resulting in limited statistical power. Therefore, we were not able to perform a multivariate analysis with the factors associated with HCC recurrence in univariate analyses. Furthermore, the utility of PIVKA-II is restricted to patients without vitamin antagonists and is not routinely measured in LT centers.

In this single-center retrospective cohort study, we demonstrated that a PIVKA-II level ≤ 90 mAU/mL can discriminate between patients with a low or higher HCC recurrence risk. Patients with low PIVKA-II, particularly those with a concomitant low AFP level or those within the Milan criteria based on explant pathology, may be considered exempt from HCC surveillance after LT. Furthermore, PIVKA-II has a stronger discriminative power to predict explant microvascular invasion compared to AFP. This suggests that PIVKA-II should be integrated in future models to identify LT candidates at high risk of HCC recurrence.

## 5. Conclusions

In conclusion, patients with low PIVKA-II levels in combination with low AFP levels or that meet the Milan criteria had 100% recurrence-free survival, potentially exempting them from post-LT HCC surveillance. PIVKA-II’s better prediction of microvascular invasion suggests its role in identifying high-risk HCC recurrence post LT.

## Figures and Tables

**Figure 1 cancers-16-00004-f001:**
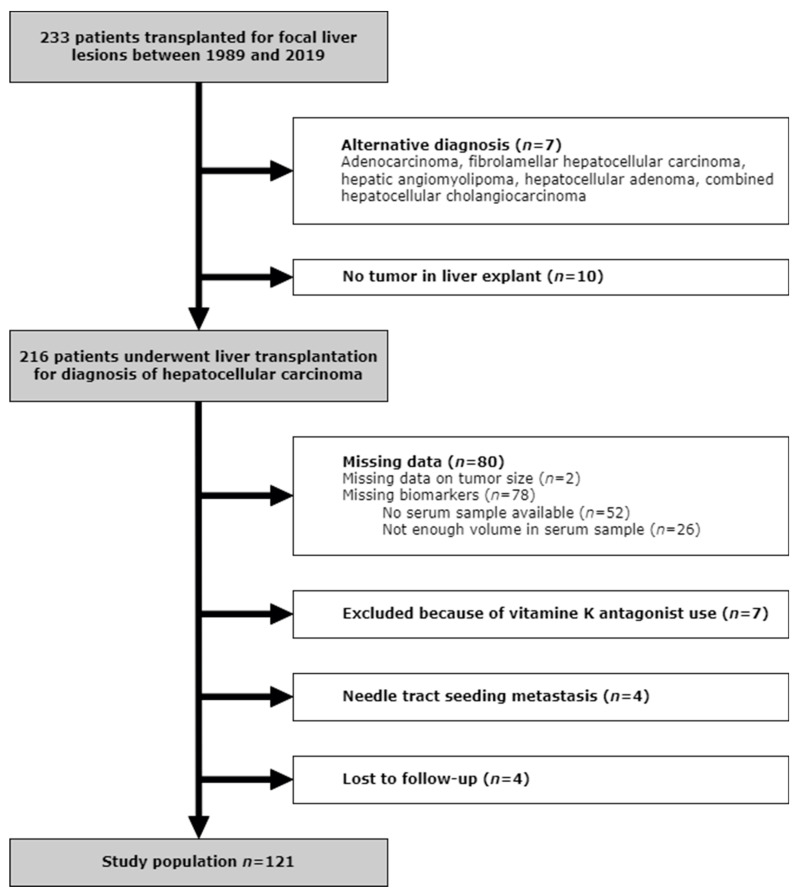
Flowchart.

**Figure 2 cancers-16-00004-f002:**
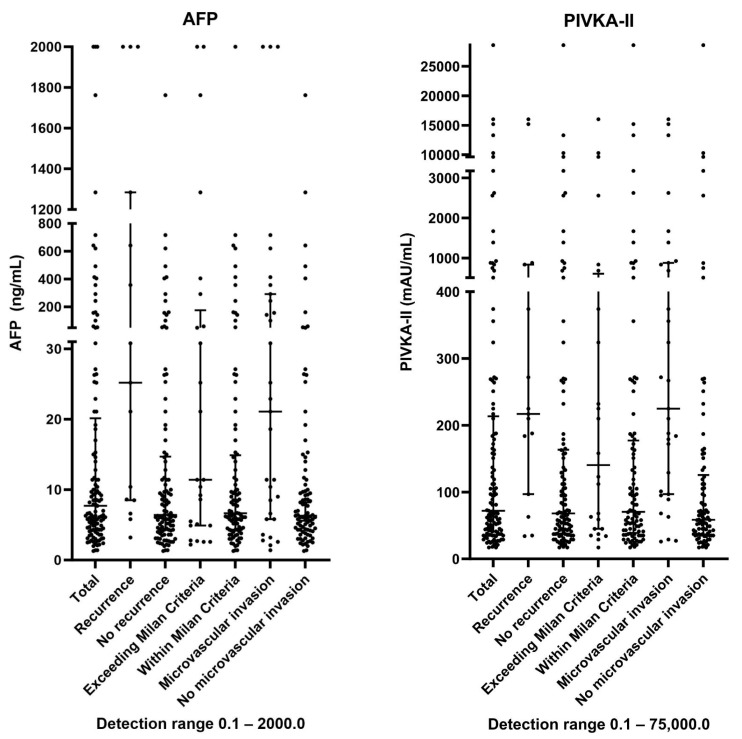
Distribution of AFP and PIVKA-II levels in different subgroups.

**Figure 3 cancers-16-00004-f003:**
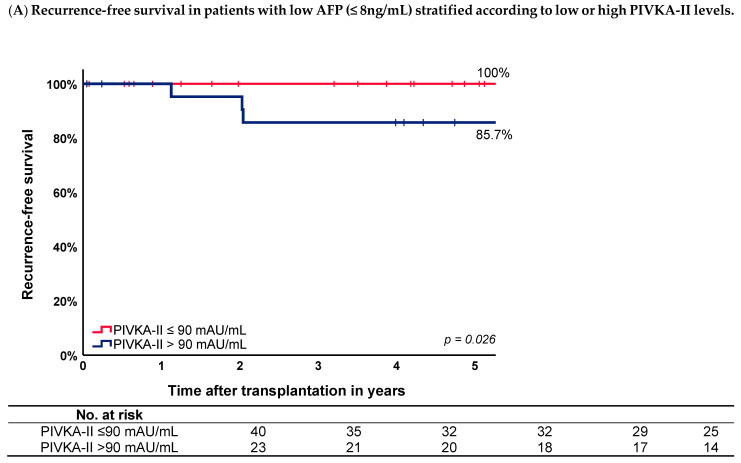
(**A**) Recurrence-free survival curves in patients with low AFP (≤8 ng/mL) stratified according to low or high PIVKA-II levels. (**B**) Recurrence-free survival curves in patients within the Milan criteria stratified according to low or high PIVKA-II levels.

**Figure 4 cancers-16-00004-f004:**
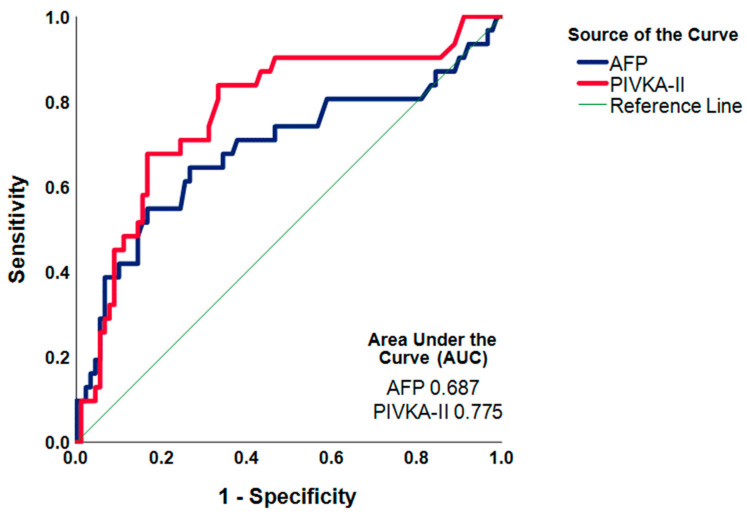
ROC curve of AFP and PIVKA-II for microvascular invasion.

**Table 1 cancers-16-00004-t001:** Patient characteristics with and without HCC recurrence.

Variable		Total Cohort (*n* = 121)	Recurrence (*n* = 15)	No Recurrence (*n* = 106)	*p*-Value
Age on LT (y)		60.7 (53.4–66.1)	59.9 (53.7–68.2)	60.8 (53.0–66.1)	0.777
Male sex		93 (76.9%)	12 (80.0%)	81 (76.4%)	1.000
Cirrhosis		118 (97.5%)	15 (100%)	103 (97.2%)	1.000
Etiology					
	Viral	53 (43.8%)	9 (60.0%)	44 (41.5%)	0.266
	NAFLD	15 (12.4%)	1 (6.7%)	14 (13.2%)	0.691
	Alcoholic	27 (22.3%)	4 (26.7%)	23 (21.7%)	0.741
	Cryptogenic and others	26 (21.5%)	1 (6.7%)	25 (23.6%)	0.188
Treatment with curative intention pre-LT		60 (49.2%)	6 (40.0%)	54 (50.9%)	0.583
Locoregional therapy pre-LT		32 (26.4%)	7 (46.7%)	25 (23.6%)	0.068
Laboratory MELD-score at time of LT		12.2 (9.3–16.2)	9.4 (8.6–14.1)	12.5 (9.3–17.8)	0.087
Time on waiting list (mo)		7.7 (3.7–11.1)	8.1 (4.9–15.3)	7.5 (3.4–10.9)	0.200
Within MC criteria based on pathology		96 (79.3%)	7 (46.7%)	89 (84.0%)	0.003
No. of viable tumors					
	No viable tumor	19 (15.7%)	0	19 (17.9%)	0.124
	1 tumor	49 (40.5%)	5 (33.3%)	44 (41.5%)	0.589
	2 tumors	24 (19.8%)	3 (20.0%)	21 (19.8%)	1.000
	3 tumors	13 (10.7%)	3 (20.0%)	10 (9.4%)	0.204
	4 or more tumors	16 (13.2%)	4 (26.7%)	12 (11.3%)	0.112
Greatest viable tumor (cm)		1.4 (0.7–2.3)	3.0 (2.0–4.0)	1.2 (0.6–2.0)	<0.001
Microvasculair invasion		31 (25.6%)	12 (80.0%)	19 (17.9%)	<0.001
Differentiation grade of the tumor					
	Unknown	2 (1.7%)	1 (0.9%)	1 (6.7%)	0.233
	Good	23 (19.0%)	0	23 (21.7%)	0.072
	Moderate	62 (51.2%)	9 (60.0%)	53 (50.0%)	0.584
	Poor	15 (12.4%)	5 (35.3%)	10 (9.4%)	0.021
AFP (ng/mL)		7.7 (4.4–20.2)	25.2 (8.5–1284.4)	6.4 (4.3–14.7)	0.001
PIVKA-II (mAU/mL)		72.0 (41.0–213.5)	217.0 (79.0–836.0)	68.0 (37.8–163.5)	0.005
Follow up time (mo)		65.9 (37.6–105.0)	18.2 (8.0–30.4)	73.1 (48.4–108.9)	<0.001
Death		47 (38.8%)	13 (86.7%)	34 (32.1%)	<0.001

Abbreviations: HCC, hepatocellular carcinoma; LT, liver transplantation; MC, Milan Criteria; MELD, Model For End-Stage Liver Disease. Data are reported as number (percentage) of patients and compared using the Fisher’s Exact Test, or reported as median (interquartile range) and compared using the Mann-Whitney U test.

**Table 2 cancers-16-00004-t002:** A: AFP and PIVKA-II levels in patients within the Milan Criteria at explant with and without HCC recurrence. B: AFP and PIVKA-II levels in patients with and without microvascular invasion at explant.

**A. AFP and PIVKA-II compared in patients within Milan Criteria**			
Biomarker	Recurrence (*n* = 7)	No Recurrence (*n* = 89)	*p*-Value
AFP (ng/mL)	8.5 (5.8–641.7)	6.4 (4.3–14.3)	0.174
PIVKA-II (mAU/mL)	217.0 (184.4–879.0)	65.0 (37.0–154.0)	0.003
**B. AFP and PIVKA-II compared in patients with microvascular invasion**			
Biomarker	Microvascular invasion (*n* = 31)	No microvascular invasion (*n* = 90)	*p*-value
AFP (ng/mL)	21.1 (5.8–292.1)	6.3 (4.3–11.5)	0.002
PIVKA-II (mAU/mL)	225.0 (97.0–879.0)	58.5 (36.8–125.5)	>0.001

Abbreviations: AFP, alpha-fetoprotein; PIVKA-II, protein induced by vitamin K deficiency or antagonist-II. Data are reported as median (interquartile range) and compared using the Mann–Whitney U test.

**Table 3 cancers-16-00004-t003:** Microvascular invasion on explant stratified according to low/high AFP (cutoff ≤ 8 ng/mL) and PIVKA (cutoff ≤ 90 mAU/mL).

		Total	Low AFP & Low PIVKA-II	High AFP & Low PIVKA-II	Low AFP & High PIVKA-II	High AFP & High PIVKA-II	*p*-Value
Microvascular invasion on explant (n, %)	no	90	40 (44.4%)	20 (22.2%)	14 (15.6%)	16 (17.8%)	<0.001
	yes	32	0 (0.0%)	6 (18.8%)	9 (28.1%)	17 (53.1%)	
	total	122	40	26	23	33	

Abbreviations: AFP, alpha-fetoprotein; PIVKA-II, protein induced by vitamin K deficiency or antagonist-II. Data are compared using chi-square test.

## Data Availability

Due to the nature of this research, the participants of this study did not agree to their data being shared publicly, and so supporting data is not available.

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
