# Peer review of "The Role of PIVKA-II as a Predictor of Early Hepatocellular Carcinoma Recurrence-Free Survival after Liver Transplantation in a Low Alpha-Fetoprotein Population"

_cancers, 2023, doi:10.3390/cancers16010004_

Round 1

Reviewer 1 Report

Comments and Suggestions for Authors

In this paper Devillers et al described the role of PIVKA-II as a predictor of HCC recurrence after liver transplantation. The study is very interesting but it needs the following clarification: 

  1. Do you confirm that surveillance of HCC recurrence is done using only clinical visits, laboratory tests and a yearly chest X-ray and electrocardiogram and not using a per protocol ultrasound or CT scans? 
  2. HCC recurrence is often silent in its early stages. How did you make a “clinical” diagnosis of recurrence? If you used the AFP abnormality as an indicator of recurrence, you probably missed those without AFP increase. 
  3. In this study, you analyze patients transplanted for HCC over a period of 30 years. How is it possible that the average follow-up is only 32.9 months? And why is the follow-up not updated at least in December 2022? 
  4. Can you specify the value of PIVKA II in the 19 patients without viable malignancy at the patological evaluation of the native liver? 
  5. To make the paper even more interesting, it would be useful to evaluate, in your cohort, the accuracy of the RETREAT with and without adding the value of PIVKA II alone and in combination with AFP 
  6. Please check some typo errors: in Table 1, the number of cirrhosis exceeds the number of patients and adding the number of patients with relapse to that of patients without relapse, 2 patients are missing. In the text the viral etiology is present in 41 patients (33.6%), while, in the Table 1 viral etiology is in 53 patients (43,4%). In Table 1, please correct the death rate (not 82.8% but 32,8%). In Table 2, microvascular invasion should be replaced with “Milan Criteria in” and “Milan citeria out”, but the “Milan in” patients would still be 96 and not 90, if what is described in point A is correct.  

Reviewer 2 Report

Comments and Suggestions for Authors

The topic is very important and suitable for the magazine. The work was done well and the results were well described. In my opinion, the article can be published with minor additions, about which I have several suggestions.

In the introduction, it makes sense to give a little more information about PIVKA-II and its role in normal and cancer cells.

In addition, as the authors rightly noted, PIVKA-II-related studies have been conducted in Eastern countries for many years. In this regard, it is probably necessary to expand the discussion part with a more detailed analysis of the results obtained for the Eastern and Western populations.

Round 2

Reviewer 1 Report

Comments and Suggestions for Authors

The authors answered to all my concerns

Author Response

As there were no further concerns, no further reply is necessary.